# Microscopic Investigation for Experimental Study on Transverse Cracking of Ti-Nb Containing Micro-Alloyed Steels

**DOI:** 10.3390/ma17040900

**Published:** 2024-02-15

**Authors:** Serkan Turan, Hossam Shafy, Heinz Palkowski

**Affiliations:** 1Institute of Metallurgy, Clausthal University of Technology, 38678 Clausthal-Zellerfeld, Germany; heinz.palkowski@tu-clausthal.de; 2HS-Metallurgical Solutions, Rudower Chaussee 5, 12489 Berlin, Germany

**Keywords:** continuous casting, hot ductility, micro-alloyed steels, transverse cracking, intergranular cracking, deformation-induced ferrite

## Abstract

The influence of Ti on the behavior of hot ductility was examined in four different Ti-containing micro-alloyed steels with a constant content of Nb. Thermomechanical investigations using a dilatometer were carried out to simulate the conditions during casting and cooling in the strand of a continuous caster with temperatures in the range of 650–1100 °C, strain rates of 0.01 s^−1^ and 0.001 s^−1^, and reheating rates between 60 and 180 Kmin^−1^. To understand the fracture mechanism, optical (LOM) and scanning electron microscopy (SEM), elemental analysis via energy dispersive X-ray spectroscopy (EDX), MatCalc “Scheil–Guilliver” calculations, and precipitation kinetics calculations were carried out for the critical conditions, showing low hot ductility between Ar_3_ and Ae_3_ temperatures and a brittle to ductile transition temperature at 900 °C. The existence of TiNb(CN), thin ferrite formation, and grain boundary sliding (GBs) due to limited dynamic recrystallization (DRX) has been documented and discussed. As a result, the reheating rate has no sufficient effect on the ductility. The existence of Nb-rich TiNb(CN) of sizes below ~1 μm triggers brittle fracture by increasing the frequency of micro-voids around grain boundaries. It can be stated that if the conditions in the hot ductility trough are avoided, the addition of Ti and high strain support minimize the risk of crack formation.

## 1. Introduction

The occurrence of transverse cracks during continuous casting (CC) leads to increasing operational costs due to the time-consuming and expensive slab repairing, or—downstream—the slab or strip even being scrapped because of malfunction. Transverse cracks preferably form particularly when the strand is straightened under mechanical and thermal stresses along the curved strand [1,2,3]. A series of studies indicated that the crack formation of steel slabs depends on a number of process parameters, such as temperature, stress, casting speed, oscillation frequency of the tundish, primary and secondary cooling, etc., as well as the composition of the alloy [1,4,5,6]. The formation of carbonitrides of Ti, Nb, and V grain boundary (GB) sliding with the limited dynamic recrystallization (DRX) of austenite, austenite–ferrite transformation, and thin ferrite films around GBs are documented as the key factors affecting the hot ductility behavior, i.e., crack susceptibility [5,7,8,9,10,11,12]. It can be generally stated that the ductility decreases particularly between 700 and 1000 °C, which represents the temperature interval during the straightening operation [1]. High-carbon equivalent, high-strength micro-alloyed steels containing alloying elements such as Nb, V, Al, and Ti are prone to surface crack defects. Fine second-phase particles of carbonitrides in these steels promote the formation of micro-voids along the GBs with correspondingly intergranular micro-cracks, which results in cracking on the macro-scale [1,2,12]. Critical points affecting the hot ductility and crack formation behavior can be listed as follows:i.Temperature reversion during secondary cooling is considered as a contributing factor affecting the hot ductility behavior. During cooling in the strand, the slab cools rapidly on the outside to a minimum temperature approx. 100 K below the deformation temperature, before being reheated by the hotter inside. This can result in a clear ductility loss, particularly for steel grades containing C-Mn-Al and C-Mn-Nb [13,14].ii.The low-temperature end of the ductility trough is associated with the formation of thin ferrite along GBs around the formation temperature of the ferrite (Ar_3_). At low strain rates, the density of dislocations increases more easily in ferrite and causes the strain hardening effect, resulting in a non-uniform strain distribution at the ferrite–austenite interface. This leads to the coalescence of micro-voids as a function of time, i.e., intergranular fracture-sensitive regions, and hence poor hot ductility. Reversely, a high strain rate ensures more stable ferrite and limited time for the nucleation of micro-voids, and correspondingly better ductility. Below Ar_3_, increasing volume fractions of ferrite will lead to a more uniform strain distribution and an interlocking boundary interface between austenite and ferrite, and hence the recovery of the ductility [15,16].iii.A total recovery of ductility is possible when approximately 40% of the austenite has transferred to ferrite [10,16]. Therefore, the ferrite formation temperature (Ar_3_) can be defined as the point at which ductility begins to recover with a further decrease in temperature. Additionally, as the deformation time increases, ferrite can form even at temperatures higher than Ar_3_. This leads to the so-called “deformation-induced ferrite” (DIF) phase. The presence of DIF can inhibit the DRX and bring about the void coalescence. The presence of this phase influences the width and depth of the ductility trough over a very wide temperature range from Ar_3_ to Ae_3_ [16,17].iv.Above the equilibrium ferrite formation temperature (Ae_3_), GB sliding due to the limited DRX becomes the dominant factor that inhibits ductility recovery. DRX reduces the grain sizes and encourages the mobility of the GBs, which enables their replacement. However, in the case of limited DRX, GB sliding occurs and intergranular crack formation is triggered by brittle fine precipitates at the prior austenite GBs [1,15,18].v.Nb is one of the most detrimental elements in terms of the encouragement of cracks in steels slabs during CC, in spite of it being a crucial element for achieving simultaneous toughness and strength through grain refinement. The existence of Nb in micro-alloyed steel leads to a deeper and wider ductility trough [4,6,17]. Nb(CN) dissolves and precipitates as fine particles in the austenite grains in the temperature range of the straightening phase of CC. The strengthened matrix enables the GB sliding and retards the DRX, and hence creates a crack-susceptible microstructure under stress–strain conditions [1,5]. Furthermore, a higher level of N promotes transverse cracking in Nb-containing steels; however, this can be minimized by keeping N below 0.004% with the help of carbonitride-forming elements such as Ti and Nb [6]. Additionally, the coexistence of Ti can encourage the improvement of hot ductility by reducing the stoichiometric ratio of the detrimental Nb(CN) particles via selective compounding with N [6,19,20,21]. Patrick and Ludlow assessed that a Ti addition of 0.02–0.04 wt.% is required to reduce transverse cracking, but 0.15 wt.% Ti is required for avoiding these cracks completely. At a ratio Ti/N = 3.6, steel slabs tend to show no transverse or corner cracks [6]. Consequently, the effects of Ti on hot ductility are complex, where the size, amount, and density of TiN precipitates are decisive in affecting the behavior of Ti [22,23].

The synergistic interaction of Ti and Nb in micro–alloyed steel and their effects on the hot ductility are not yet completely understood. Therefore, in this research study, detailed hot ductility diagrams were built depending on different strain and reheating rates in the temperature range of 650–1100 °C. Accordingly, in the light of the complex material and process effects discussed above, the formation of micro-defects was studied using light optical (LOM) and scanning electron microscopy (SEM) (Carl Zeiss AG, Jena, Germany) as well as energy dispersive X-ray (EDX) (Oxford Instruments, Abingdon, UK), and MatCalc software (Version: 6.01 (rel 1.003), Institute of Materials Science and Technology—TU Wien, Wien; Austria) was used to simulate the precipitations.

## 2. Materials and Methods

Three different Ti/N ratios were investigated (represented by A1, A2, and A3) to examine the effect of Ti on the hot ductility in the presence of constant Nb and N. Additionally, alloy A0 without Ti was used as a benchmark. The Ti/N ratio in A1 was less than 3, around 3 in A2, and much higher than 3 in A3. The chemical compositions of the investigated steels are shown in Table 1. Each alloy contained 0.04% of Nb to study the interactive effect of Ti(CN) and Nb(CN). 

Alloys to be tested were molten in a vacuum induction furnace and cast in a 155 × 60 × 155 mm^3^ mold, and tensile samples were processed using electrical discharge machining (EDM) wire machining according to the dimensions specified in Figure 1c.

In addition to tensile specimens, cylindrical samples sized Ø 10 mm × 5 mm were also produced using EDM wire machining for determining the austenite–ferrite transformation temperature (Ar_3_). 

The experimental procedure was carried out using a Baehr dilatometer (DIL 805 T/A, TA instruments, New Castle, DE, USA) (see Figure 1a,b). The first tests were performed in the quenching mode of dilatometry with a slow cooling rate of 0.015 Ks^−1^ to define the austenite–ferrite transformation temperature (Ar_3_) representing the low-temperature end of the ductility trough. 

To measure the hot ductility, a hot tensile test (tensile mode of dilatometry) simulating the straightening process in CC was implemented with different parameters in an inert nitrogen atmosphere. Related thermomechanical treatments followed the steps (see Figure 2): the tensile samples were heated up at a heating rate of 1000 Kmin^−1^ to the austenitization temperature T_aus_ = 1320 °C (~30 K below the nil-strength temperature (NST)), which was determined previously using a simulator of thermomechanical metallurgical states Gleeble 1500D (Dynamic Systems Inc., Poestenkill, NY, USA) at the University of Ljubljana (Faculty of Materials and Metallurgy Department of Materials and Metallurgy). After this, the sample was kept for 10 min to dissolve existing precipitates and ensure homogenous temperature distribution. This generated an initial microstructure before the straightening process resembling CC. Then, the samples were cooled down at the rate of 600 Kmin^−1^ to a temperature 100 K below the test temperature. After this step, the samples were reheated to the test temperature with a heating rate v of 60 or 180 Kmin^−1^, alternatively, to simulate the temperature reversion process of secondary cooling during CC. To prepare for deformation, samples were kept for 3 min at the test temperature to ensure a homogenous temperature distribution. Finally, tensile tests were carried out with strain rates of έ_1_ = 0.001 s^−1^ and έ_2_ = 0.01 s^−1^ and test temperatures ranging from 650 °C to 1100 °C until complete failure, followed by rapid cooling to room temperature to freeze the microstructure for microscopic analysis. Hot ductility was measured by the flow stress–deformation rate curves obtained at the end of the experiments representing the reduction of area (RA). The thermomechanical treatment is given schematically in Figure 2. 

## 3. Results and Discussion

### 3.1. Calculation of Precipitates via MatCalc Software

The Scheil–Gulliver model was used to calculate the solidification behavior of the primary precipitates of TiN and NbC under the equilibrium condition. This provided the information about precipitates still existing in the matrix. They can precipitate during further cooling. Back-diffusion of C and N and the peritectic reaction were also considered in the calculation. Whereas the primary precipitations are fixed, the secondary precipitates form on GBs. 

The solidification points of the alloys varied within the range of 1390–1430 °C, as can be seen in Figure 3, with a 0.01% liquid phase. Significant increases in the primary precipitates were observed at the end of solidification, and an increase in the formation-frequency of TiNb(CN) in A3 where Ti/N is apparently higher than 3 can be expected, because the phase fraction of NbC tends to increase simultaneously with the high amount of TiN (see Figure 3c). 

Fine secondary precipitates inhibit the DRX, and their size, amount, and distribution play a crucial role in the thermomechanical behaviour of the steels [15]. In addition to primary solidification, the features of secondary particles have been simulated using a precipitation kinetics model via MatCalc, too, and they are discussed in the Fracture Mechanism section. In these models for the individual themomechanical cycle, dislocation (dln) and grain boundary (GB) kinetics have been considered by defining the nucleation of secondary particles.

### 3.2. Experimental Work

#### Ar_3_ and Ae_3_ Measurements

The Ar_3_ temperatures, determined using the forming simulator as described in the previous section, and the Ae_3_ temperatures, calculated using MatCalc software, are listed in Table 2. The Ar_3_ values are the averages of a minimum of two measurements each.

### 3.3. Thermomechanical Tests

A minimum of two experiments were performed for each thermomechanical cycle. The experimental results are given with their error bars in Figure 4. General results valid for all steel grades are as follows:

The hot ductility results did not significantly alter with the change in the reheating rate (60 and 180 Kmin^−1^), particularly at a low strain rate (έ = 0.001 s^−1^). On the other hand, the change in the reheating rate in combination with a rapid strain rate (έ = 0.01 s^−1^) fluctuated the hot ductility results with altering temperature; however, no clear regularity between the temperature and reheating rate could be stated (see Figure 4b,d). The low-temperature end of the ductility trough was the test temperature of 700 °C representing an approximate Ar_3_ for each process parameter and alloy (see Table 2). The hot ductility recovered below 700 °C (see the results for 650 °C in Figure 4c,d) was due to the increasing volume fraction of ferrite on GBs, which had not been discussed in this work. A crucial result to emerge from the experiments was that the plateau of the hot ductility trough was spread between Ar_3_ and Ae_3_ temperatures at the deformation rate of έ = 0.001 s^−1^. On the other hand, an increase in the strain rate (έ = 0.01 s^−1^) improved the hot ductility remarkably in the range of the ductility trough. This improving effect of a high strain rate was prone to be minimized above 900 °C. Subsequently, the hot ductility began to increase above ~Ae_3_ for each strain rate and reheating rate. A temperature of 900 °C represented the high-temperature end of the hot ductility, i.e., the brittle to ductile transition temperature.

Investigated steels showed different hot ductility behaviors in some points as follows:

Increasing the Ti content tended to improve the hot ductility under conditions of έ = 0.01 s^−1^ between 900 °C and 1100 °C (see Figure 4b,d). The most conspicuous observation to emerge from the results’ comparison was that, compared to A0 at 900 °C, the Ti-containing steels (A1, A2, A3) revealed better hot ductility in all other parameters. Furthermore, A2 and A3 were prone to reveal slightly better hot ductility at temperatures of 800 °C and 850 °C with έ = 0.001 s^−1^ (see Figure 4a,c). A2 showed a distinctly higher hot ductility at 800 °C and 850 °C with έ = 0.01 s^−1^ (see Figure 4b,d). On the other hand, at 700 °C and under a high strain rate (0.01 s^−1^), Ti-containing steels showed poorer hot ductility than A0 (see Figure 4b,d). Curiously, uniquely to steel A2, a slight decrease in the hot ductility from 1000 °C to 1100 °C with έ = 0.001 s^−1^ was observed, as labelled in Figure 4a,c. On the other hand, the hot ductility of the other steel grades (A0, A1, A3) tended to increase at each strain rate and reheating rate between 1000 °C and 1100 °C. In this work, the reason for this hot ductility trend was not studied, because RA values were above the ductility trough; in other words, there was a lower crack probability.

### 3.4. Fracture Mechanism

LOM, SEM, and EDX analyses were performed for the purpose of understanding the microscopic fracture mechanism giving rise to micro-cracks. The metallographic investigations focused specifically on samples with a hot ductility approximately below 40%, where the risk of transverse cracks increases according to the study of Minz et al. [1], and 900 °C, defined as the critical transition temperature between low and high hot ductility regions. The ductility values given in the microscopic images represents the average values.

The samples used for ferrite film and carbo-nitride detection were subjected to rapid cooling at the end of the thermomechanical treatment to freeze the microstructure at test temperature. Furthermore, due to the negligible effect of the reheating rate, only temperature and strain-rate values were the main parameters for the selection of samples to be investigated microscopically.

The hot ductility was prone to decrease by increasing the Ti content at 700 °C with the strain rate of 0.01 s^−1^, as can be seen in Figure 4b,d. Each investigated steel corresponding to low hot ductility showed intergranular fracture surfaces at 700 °C (near Ar_3_) and έ = 0.001 s^−1^. While at έ = 0.001 s^−1^, the hot ductility was similar for each alloy, A0 showed an improved recovery (RA = 46%) performance at a strain rate of έ = 0.01 s^−1^ and v = 60 Kmin^−1^. At this strain rate, while A0 had a trans-crystallized structure, the surface of Ti-containing steels had explicitly intergranular fracture morphologies (see Figure 5).

The dominant factor around Ar_3_ causing the intergranular fracture morphology is associated with the ferrite nucleation and the corresponding pro-eutectoid ferrite film that formed on the boundaries of previous austenite grains (see Figure 6). The fracture mechanism can be categorized as ductile at the microscopic scale and brittle at the macroscopic scale. A slowly deformed microstructure with a strain rate of 0.001 s^−1^ at this temperature leads to an increase in stored dislocations in thin ferrite phase [15,24]. This may cause the formation of micro-voids along the thin ferrite film formed on GBs, and hence the coalescence of these voids at a low strain rate. The growth of these micro-voids and the occurrence of the intergranular cracks along GBs were observed for each alloy, as can be seen in Figure 6. The difference in the thickness of the ferrite film at 700 °C is slightly correlated with the results of the experiments for the measurement of Ar_3_ as listed in Table 2. The ferrite layer was thicker in A1 due to the higher Ar_3_ (see Table 2). The steel with the thinnest ferrite film (A2) had the lowest ductility value (RA = 15%). This was more pronounced at 180 Kmin^−1^. While A2 had the lowest Ar_3_, A1 exhibited the highest one, as shown in Table 2.

It was often observed in A0, A1, and A2 at έ = 0.001 s^−1^ that the morphology of the ferrite film was acicular (AF) in many cases and grew from the allotriomorphic ferrite into the matrix. This structure reduces the probability of crack propagation in the periphery of this field. However, polygonal ferrite (PF) plates, albeit in different sizes, were also observed in all steel grades. In general, the formation of micro-voids and intergranular cracks has been observed around polygonal ferrite plates (see Figure 6). Polygonal ferrite morphology is more likely to trigger intergranular crack formation.

Even at 700 °C, the precipitates in A3 were often microscopically (SEM) visible. With Ti/N > 3 (A3), a high amount of TiNb(CN) suppresses noteworthy recovery in the hot ductility even at 700 °C, 0.01 s^−1^. An overview of the microstructures revealed that micro-cracks were mainly based on the existence of TiNb(CN) on GBs (see Figure 7a). According to the results of MatCalc calculation for the phase fraction (%) (NbC~3.5 × 10^−5^, TiN~2 × 10^−5^) and EDX analysis, small TiNb(CN) particles tend to have a high amount of NbC (see Figure 7b,d). In this manner, the high TiN formation also increases the possibility of the formation of TiNb(CN) complex particles. Moreover, increasing the phase fraction of high Nb-containing TiNb(CN) precipitates, particularly at triple joints of grains, creates crack-susceptible microstructures. The crack starts directly at these GBs, as indicated in Figure 7c, especially if the precipitates are smaller than 1 μm.

It can also be assessed that the lowest έ = 0.001 s^−1^ extended the formation range of the ferrite film above Ar_3_, as theoretically discussed in the introduction section. In the range of the low-ductility plateau (see Figure 4), it was possible to detect various types of ferrite morphologies such as acicular, polygonal, and longitudional ferrite (see Figure 8). Additionally, in A3, the large primary TiNb(CN) particles became more frequently detectable and located on GBs and the matrix, as seen in Figure 8b.

SEM examination of A2 proved that the TiNb(CN) precipitates with a mean diameter ~<100 nm accumulated along the secondary GB together with a columnar precipitate TiNb(CN) particle with a size of ~1 μm. The micro-void that initiated around this primary TiNb(CN) elongated along the piled-up secondary TiNb(CN) particles (see Figure 9). In fact, these particles caused partial discontinuties in the microstructure, despite the large amount of relatively ductile TiN particles. The tendency for microstructural defects to appear, especially at sizes of TiNb(CN) < 100 nm, increased. This can be attributed to the fact that the NbC ratio is prone to increase in complex TiNb(CN) as it becomes finer.

A similar precipitate distribution to the one in A2 was more frequent in A3. Besides the frequency, the size of the polygonal primary precipitates of TiNb(CN) even exceeded 2 μm. The secondary TiNb(CN) directly near a micro-defect region was a typical observation on the condition that a sufficient amount of secondary precipitates was activated. These coarse primary precipitates were rich in Ti. The locations of these particles can be defined as quasi-C-poor regions. This caused C-rich zones directly around the primary precipitates, and correspondingly encouraged the selective compounding of carbon atoms with Ti and Nb during the formation of secondary precipitations. The accumulation of these small secondary C-rich TiNb(CN) precipitates at the location of GBs led frequently to the creation of micro-defects, as seen in Figure 10. The mean diameter of the detected secondary precipitates with different morphologies (polygonal, columnar) varied within the range of 100–600 nm in A3.

A temperature of 850 °C was the temperature at which precipitates were the dominant factor in microcrack formation, and their phase fraction increased significantly. 

The calculation of precipitation kinetics using MatCalc was performed for the necessary parameters at 850 °C. The results of these calculations reveal the mean diameter of individual secondary NbC and TiN in the range of 2–20 nm. According to the SEM and EDX results, secondary precipitates were distributed as complex TiNb(CN) as larger particles than the simulation suggested. The mean diameter size of crack-inducing particles varied between approximately 80 nm and 1 μm. The accumulated large amount of secondary TiNb(CN) on prior austenite GBs inhibits the DRX [1].

Also in this study, it was detected that GBs moved directly along interfaces on which brittle TiNb(CN) were settled and microscopically and macroscopically brittle fractures occurred under the conditions of a long deformation time with the strain rate of 0.001 s^−1^. NbC precipitates could not be detected by SEM and EDX in the A0 alloy due to the relatively small fraction and extremely small size of the NbC in the steel. However, fine Nb(CN) precipitates can be considered as a crucial factor leading to the worst RA in A0 (RA = 27%) at 850 °C, 0.001 s^−1^, and 180 Kmin^−1^. On the other hand, the precipitation of NbC in a complex carbonitride with Ti(CN) in A1, A2, and A3 increased the size of precipitates and made them more likely to be visible. As shown in the MatCalc simulation in Figure 11, for the A1 steel, the phase fraction of the secondary NbC exceeded even the primary precipitates, i.e., Nb was dominant in TiNb(CN). 

This result is in accordance with the SEM and EDX investigations for A1, as given in Figure 12. These brittle particles piled up along the GBs and triggered the formation of micro-cracks under deformation with simultaneous GB sliding. The diameters of the detected individual particles next to the micro-defects differed from 100 to 800 nm, and they were often of the coalesced type (see Figure 12). Thus, it was not possible to describe a reliable size distribution. The ferrite film on GBs could not be observed anymore above Ae_3_ (~800 °C) and the hot ductility with the strain rate of 0.001 s^−1^ could not exceed above ~40% because of the insufficient DRX and grain boundary sliding, and hence crack were formed. The presence of fine secondary precipitaties of Nb(CN) and TiNb(CN) on the triple joints of grains played a crucial role in this behaviour at 850 °C and a strain rate έ = 0.001 s^−1^ (see Figure 12).

As shown in the MatCalc simulations in Figure 11 for steels A2 and A3, the phase fraction of the primary and secondary particles of both TiN and NbC increased; nevertheless, TiN became clearly dominant if Ti/N > 3. The Ti content increased the hot ductility slightly up to the ratio of Ti/N ~3 (RA (A0, A1, A2) = 27%, 31%, 34%, respectively). On the other hand, as stated previously, increasing Ti/N > 3 promoted the formation of NbC in the complex TiNb(CN) simultaneously and inhibited the continuous recovery of RA (RA (A3) = 30%) because of the aformentioned detrimental effects. The diameter of particles detected next to the micro-defects in A2 and A3 differed in the ranges of approx. 40–600 nm and 20 nm−1 μm, respectively. Interestingly, the TiNb(CN) particles along the secondary GB were predominantly Ti-rich particles. The GB sliding occurred on the micro-voids caused by TiNb(CN)s under deformation. The distribution of these particles and GB sliding can be observed as zig zag-formed secondary GBs in Figure 13 and lateral slip on triple points of GBs, shown in Figure 14.

According to the hot ductility results of Ti-containing steels, there was a significant recovery in the hot ductility at 900 °C. This temperature can be defined as the transition temperature form brittleness to ductility. Under low strain rates (0.001 s^−1^), the fracture surface of A0 was classified as trans-crystalline. The microstructure of A3 and A0 (without Ti) at the strain rate of 0.01 s^−1^ showed dimple coalescence and ductile striations (see Figure 15). The intensity of ductile striations and dimple coalescence increases with a higher strain rate (0.01 s^−1^). The apparent dimple structure with nearby micro-voids of the Ti-containing A3 steel was common in improved ductility values at 900 °C (see Figure 15).

As seen in Figure 16, the fracture view remained dominantly intergranular, which means that the onset of DRX was not sufficiently activated at this deformation rate. Furthermore, the relatively lower ductility of steel A0 at 900 °C, 0.001 s^−1^ (for 60 Kmin^−1^ and 180 Kmin^−1^, RA = 37% and 38%, respectively) compared to Ti-containing steels can be explained as due to thin NbC precipitates leading to micro-defects. These micro-defects occurred due to an increase in the number of secondary TiNb(CN)s, and thus an increase in the number of micro-voids between the particles and the GBs, leading to micro-void coalescence. According to MatCalc calculations, the phase fraction of secondary NbC precipitates increased significantly during deformation at the test temperature (see Figure 16). The diameter of NbC particles varies approximately from < 50 nm towards 500 μm. It was also found that NbC particles differ from various morphologies such as angular, columnar, and spherical (see Figure 16).

## 4. Conclusions

The effect of Ti in Nb-containing steels on the formation of transverse cracks has been examined with thermomechanical treatments simulating the process conditions during continuous casting. The roles of ferrite formation, precipitation kinetics, and GB sliding in micro-defect formation were examined using microscopic analysis and MatCalc simulations. Our main findings can be summarized as follows:(1)A slow strain rate (0.001 s^−1^) broadened the hot ductility trough between nearly 700 °C and 800 °C by inducing the DIF for each steel grade.(2)For each steel grade, an increase in the reheating rate at some points resulted in irregular changes in hot ductility at the strain rate of 0.01 s^−1^; however, these changes were not statistically remarkable. Furthermore, almost no change was recorded at the slower strain rate of 0.001 s^−1^.(3)For each steel grade, an improvement in the range of ductility trough (between Ar_3_ and Ae_3_) from 15–30% to 30–50% was determined by an increase in the strain rate from 0.001 s^−1^ to 0.01 s^−1^. Thin ferrite formation was the main factor leading to micro-cracks induced by micro-void coalescence between the Ar_3_ and Ae_3_ temperatures, i.e., the hot ductility trough. In the range of the presence of a thin ferrite film, microcracks at GBs were more likely to be initiated with polygonal morphology than acicular or others. The hot ductility at 700 °C with a strain rate of 0.01 s^−1^ was prone to be lower in the Ti-containing steels compared to the non-Ti-containing steel (A0). This phenomenon can be attributed to the increasing frequency of TiNb(CN) carbonitrides in the presence of Ti, as clearly observed in the microstructure of steel A3.(4)Between 800 °C and 900 °C, the accumulation of NbC and TiN as the complex TiNb(CN) precipitates on the joints of grains caused micro-defects and inhibited the DRX, and correspondingly suppressed sufficient recovery of ductility. The precipitates below the diameter of <1 μm were generally observed as the crack initiators. On the other hand, the presence of Ti began to improve the hot ductility at a temperature of 800 °C (~Ae_3_). At this temperature, the steel grade A2 with Ti/N ~3 was unique, with values above 40% at 0.01 s^−1^ for both reheating rates of 60 and 180 K/min. Moreover, it was obvious that Ti content was favorable to ductility improvement at 900 °C even at strain rates of 0.001 s^−1^. Particularly with the ratio of Ti/N > 3 (steel A3), the high fraction of Ti promoted the precipitation of TiNb(CN) and encouraged crack-sensitive regions along GBs.

## Figures and Tables

**Figure 1 materials-17-00900-f001:**
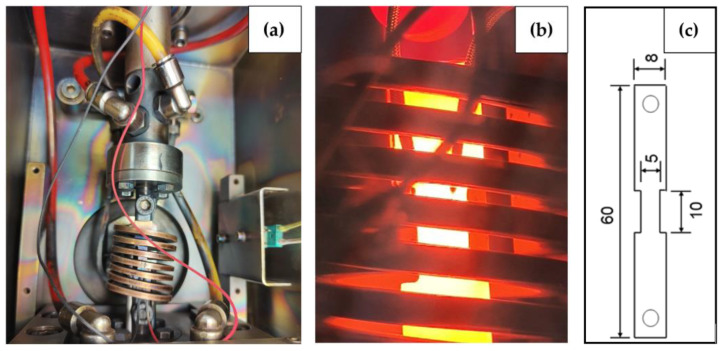
DIL 805 T/A hot tensile testing system: (**a**) installation of tensile test; (**b**) thermomechanical treatment; (**c**) sample dimensions in mm (1.8 mm thickness).

**Figure 2 materials-17-00900-f002:**
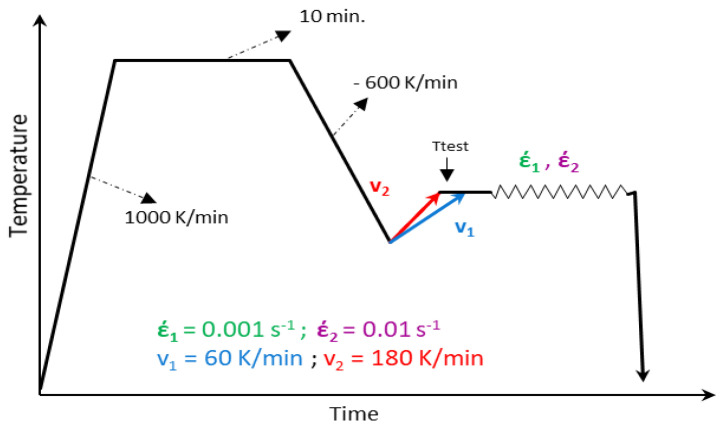
Thermomechanical cycle used.

**Figure 3 materials-17-00900-f003:**
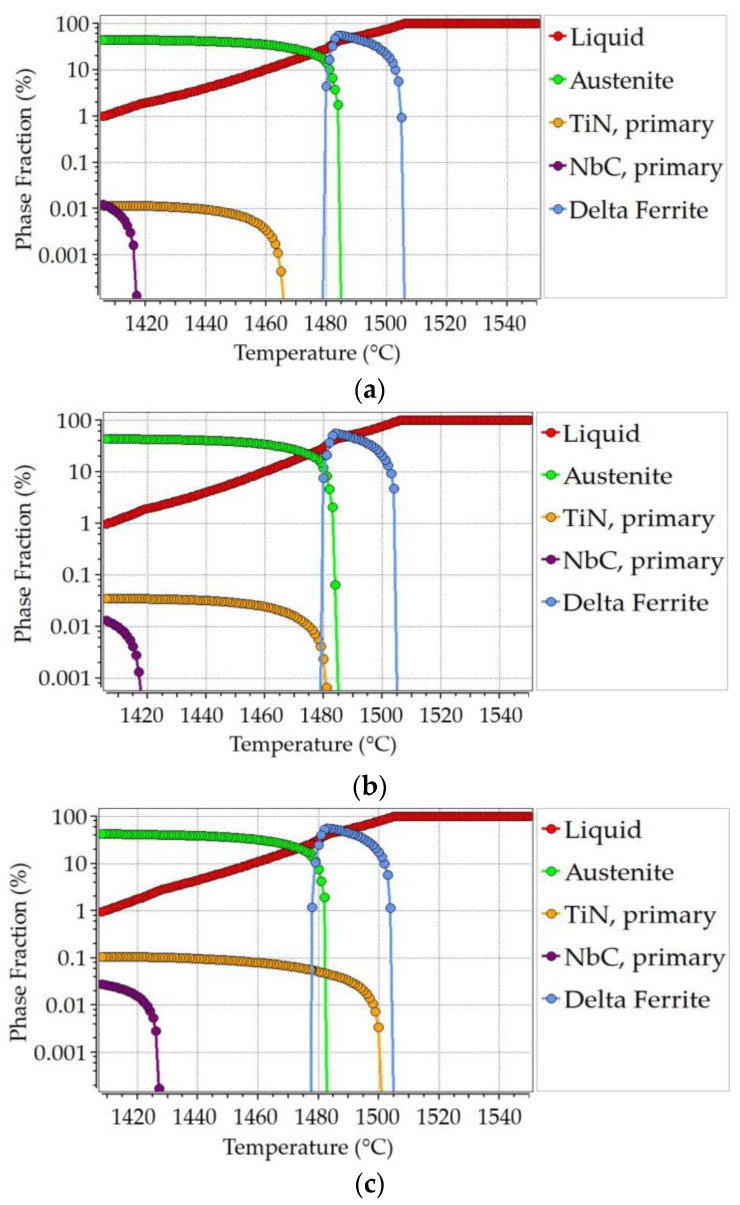
Scheil–Gulliver calculations for Ti-Nb-containing steels: (**a**) A1; (**b**) A2; (**c**) A3.

**Figure 4 materials-17-00900-f004:**
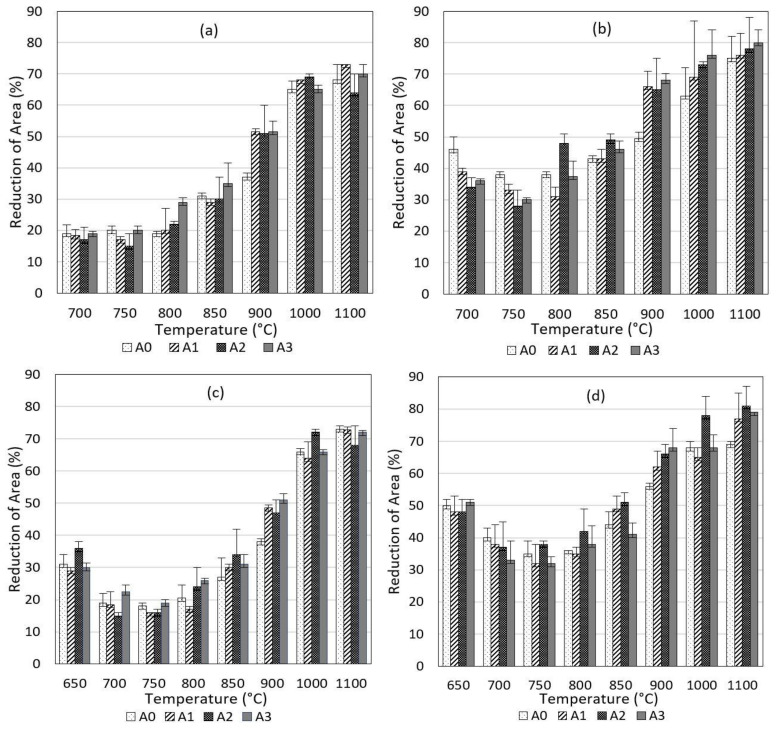
Hot ductility by reduction of area vs. temperature: (**a**) 0.001 s^−1^, 60 Kmin^−1^; (**b**) 0.01 s^−1^, 60 Kmin^−1^; (**c**) 0.001 s^−1^, 180 Kmin^−1^; (**d**) 0.01 s^−1^, 180 Kmin^−1^.

**Figure 5 materials-17-00900-f005:**
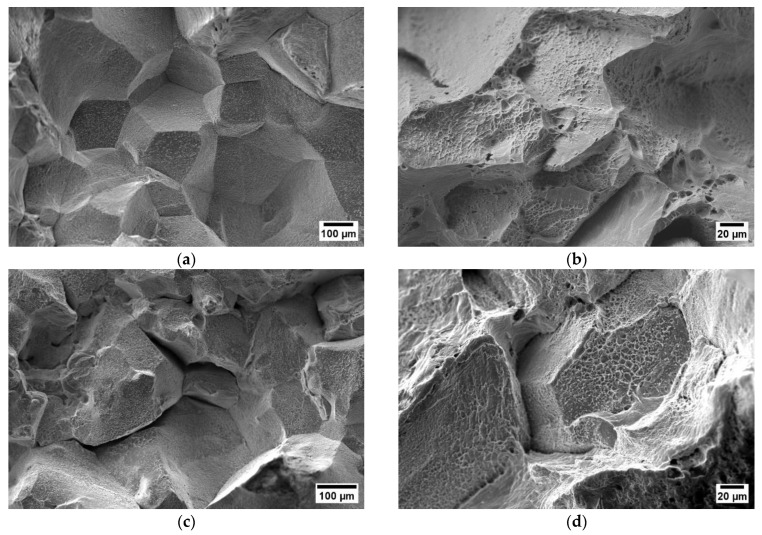
SEM micrographs of the fracture surface (parallel to the tensile force): (**a**) A0 at 700 °C, 0.001 s^−1^, 60 Kmin^−1^ (RA = 19%); (**b**) A0 at 700 °C, 0.01 s^−1^ 60 Kmin^−1^ (RA = 46%); (**c**) A3 at 700 °C, 0.001 s^−1^, 60 Kmin^−1^ (RA = 19%); (**d**) A3 at 700 °C, 0.01 s^−1^, 60 Kmin^−1^ (RA = 36%).

**Figure 6 materials-17-00900-f006:**
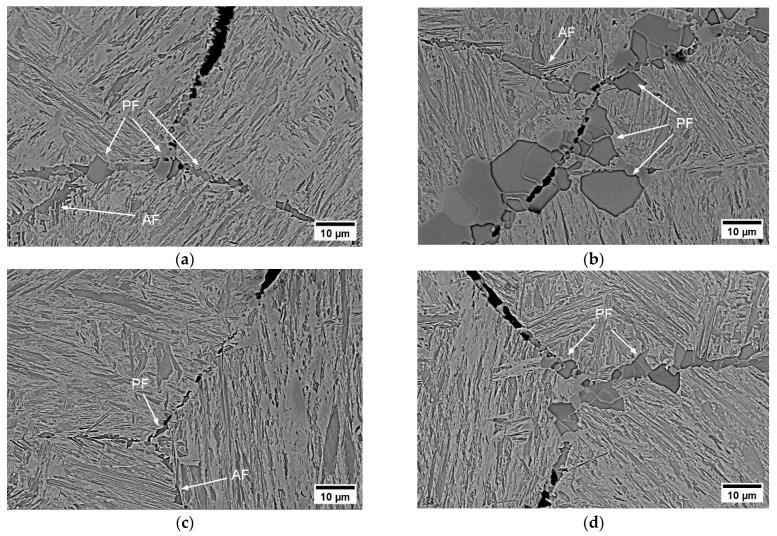
SEM micrographs near the fracture surface (perpendicular to the tensile force)—ferrite formation at GBs (AF and PF represent acicular and polygonal ferrite morphologies, respectively): (**a**) A0 (RA = 19%); (**b**) A1 (RA = 19%); (**c**) A2 (RA = 15%); (**d**) A3 (RA = 23%).

**Figure 7 materials-17-00900-f007:**
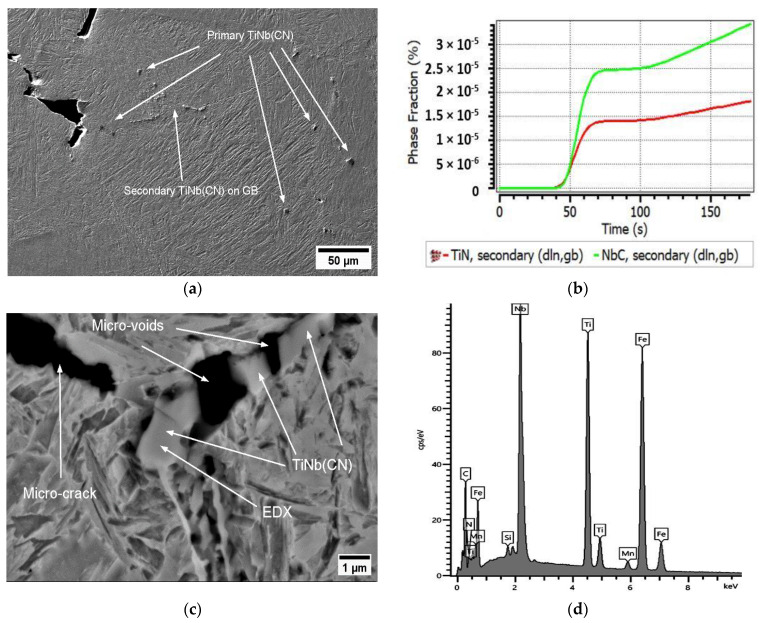
SEM micrographs near the fracture surface (perpendicular to the tensile force) of A3 at 700 °C, 0.01 s^−1^, 180 Kmin^−1^ (RA = 33%): (**a**) SEM overview image; (**b**) phase fraction of secondary precipitates (via MatCalc); (**c**) SEM image of micro-crack on GB induced by TiNb(CN); (**d**) EDX analysis.

**Figure 8 materials-17-00900-f008:**
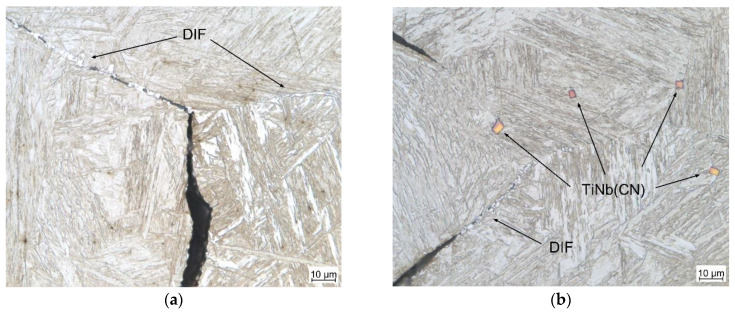
LOM micrographs near the fracture surface (perpendicular to the tensile force): DIF on GBs in (**a**) A0 (RA = 17%); (**b**) A3 (RA = 18%).

**Figure 9 materials-17-00900-f009:**
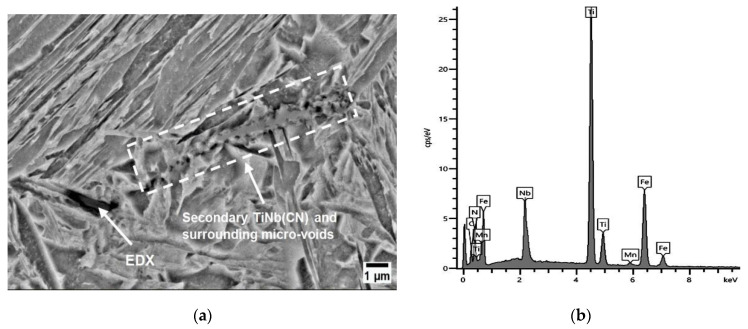
SEM micrographs of the fracture surface (perpendicular to the tensile force) for A2 at 750 °C, 0.001 s^−1^, 180 Kmin^−1^ (RA = 16%): (**a**) secondary precipitates and micro-voids; (**b**) EDX analysis.

**Figure 10 materials-17-00900-f010:**
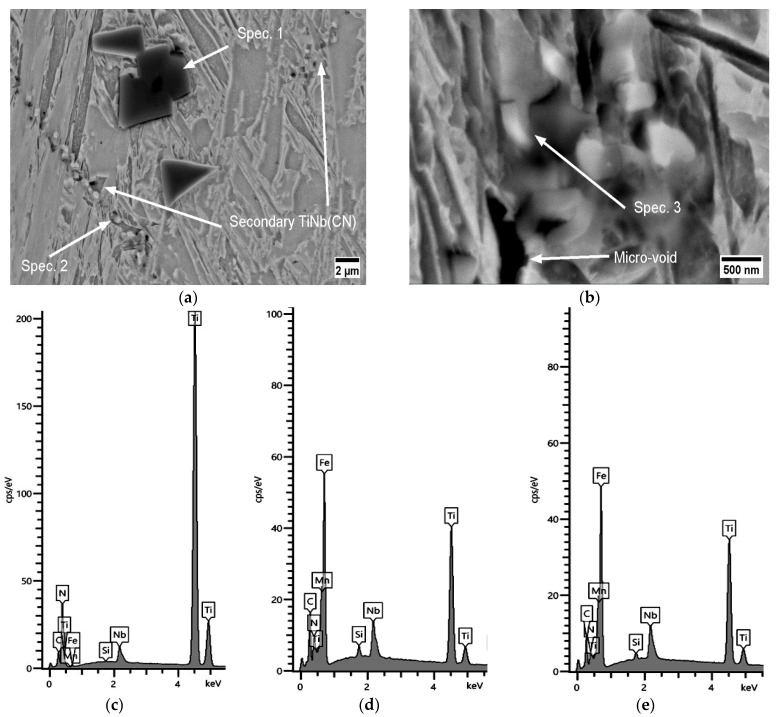
SEM micrographs near the fracture surface (perpendicular to the tensile force) for A3 at 750 °C, 0.001 s^−1^, 180 Kmin^−1^ (RA = 19%): (**a**) primary and secondary TiNb(CN) clusters; (**b**) secondary TiNb(CN) cluster and micro-void on GB; (**c**) EDX for Spec. 1; (**d**) EDX for Spec. 2; (**e**) EDX for Spec. 3.

**Figure 11 materials-17-00900-f011:**
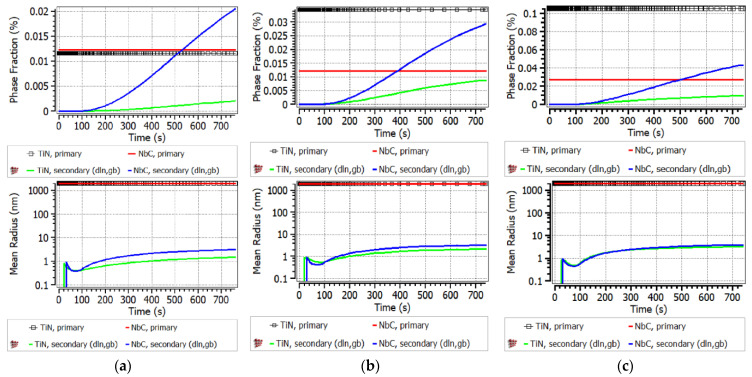
Calculations for phase fraction and mean radius via MatCalc at 850 °C, 0.001 s^−1^, 180 Kmin^−1^: primary and secondary precipitates for (**a**) A1; (**b**) A2; (**c**) A3.

**Figure 12 materials-17-00900-f012:**
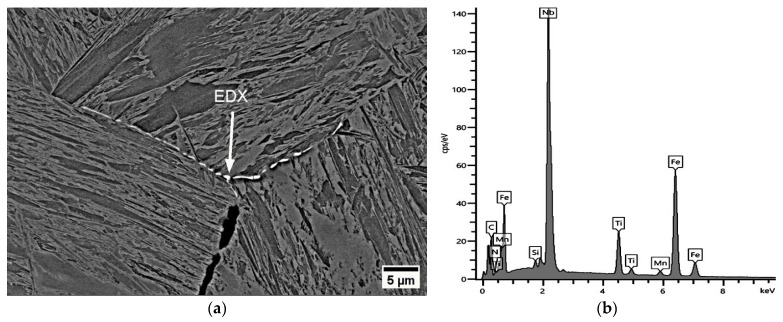
SEM micrographs near the fracture surface (perpendicular to the tensile force) for A1 at 850 °C, 0.001 s^−1^, 180 Kmin^−1^ (RA = 30%): (**a**) secondary TiNb(CN) on GB, micro-voids, GB sliding, and V-shape pile-up of secondary TiNb(CN)s; (**b**) EDX analysis.

**Figure 13 materials-17-00900-f013:**
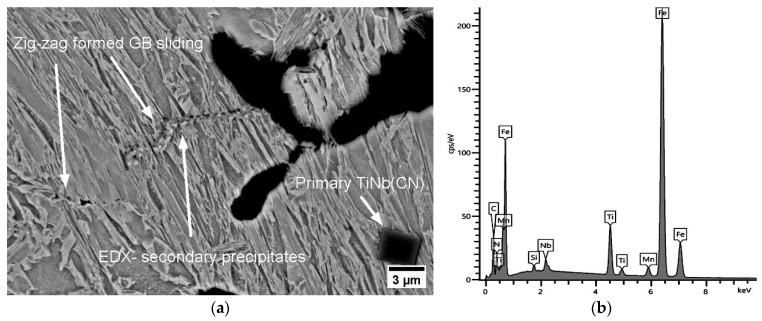
SEM micrograph near the fracture surface (perpendicular to the tensile force) for A2 at 850 °C, 0.001 s^−1^, 180 Kmin^−1^ (RA = 34%): (**a**) secondary TiNb(CN) on GB, GB sliding and micro-defect, zig zag-formed secondary TiNb(CN) on secondarily formed GB; (**b**) EDX analysis.

**Figure 14 materials-17-00900-f014:**
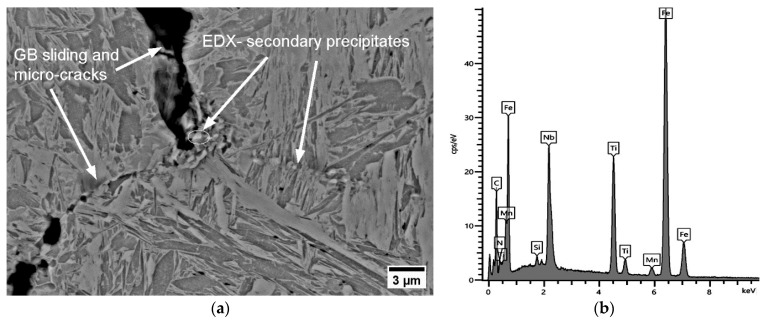
SEM micrograph near the fracture surface (perpendicular to the tensile force) of A3 at 850 °C, 0.001 s^−1^, 180 Kmin^−1^ (RA = 31%): (**a**) secondary TiNb(CN) on GB, GB sliding with lateral slip and micro-voids; (**b**) EDX analysis.

**Figure 15 materials-17-00900-f015:**
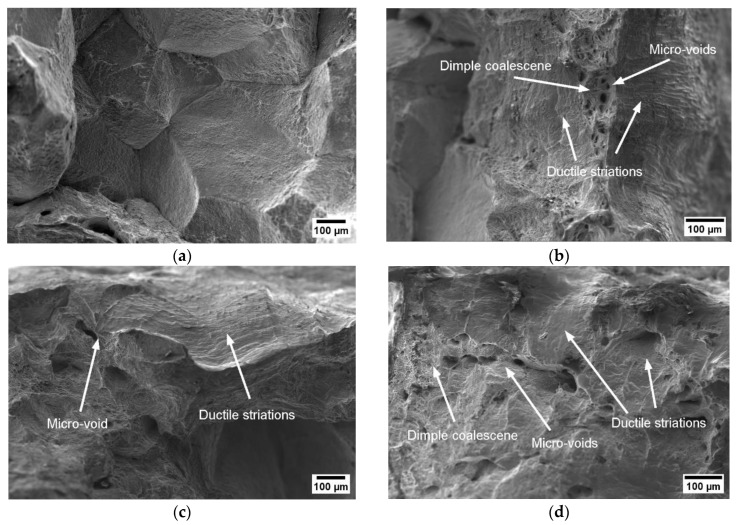
SEM micrographs of the fracture surface (parallel to the tensile force) at 900 °C, 0.001 s^−1^, and 0.01 s^−1^: (**a**) A0 at 900 °C, 0.001 s^−1^, 180 Kmin^−1^ (RA = 38%); (**b**) A0 at 900 °C, 0.01 s^−1^ 180 Kmin^−1^ (RA = 56%); (**c**) A3 at 900 °C, 0.001 s^−1^, 180 Kmin^−1^ (RA = 49%); (**d**) A3 at 900 °C, 0.01 s^−1^, 180 Kmin^−1^ (RA = 68%).

**Figure 16 materials-17-00900-f016:**
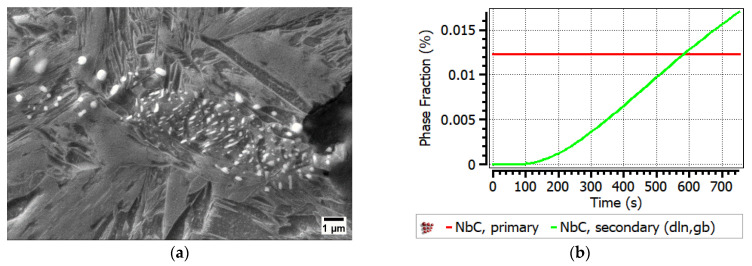
SEM micrograph near the fracture surface (perpendicular to the tensile force) for A0 at 900 °C, 0.001 s^−1^, 180 Kmin^−1^ (RA = 37%): (**a**) secondary precipitates of NbC < 500 nm dimension accumulation near micro-void; (**b**) MatCalc calculation.

**Table 1 materials-17-00900-t001:** Chemical compositions (wt%) of the material groups.

Grade	C	Si	Mn	Ti	Nb	N	Ti/N
A0	0.187	0.452	1.99	0	0.041	~0.009	-
A1	0.216	0.478	1.988	0.014	0.039	~0.008	<3
A2	0.208	0.483	1.977	0.025	0.039	~0.008	~3
A3	0.213	0.491	1.936	0.079	0.038	~0.008	>3

**Table 2 materials-17-00900-t002:** Ar_3_ and Ae_3_ values.

Steel	Ar_3_, °C	Ae_3_, °C
A0	683 (±1.0)	797
A1	701 (±4.5)	798
A2	668 (±10.0)	800
A3	685 (±1.0)	802

## Data Availability

The data presented in this study are available on request from the corresponding author.

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
