# Peer review of "Microscopic Investigation for Experimental Study on Transverse Cracking of Ti-Nb Containing Micro-Alloyed Steels"

_materials, 2024, doi:10.3390/ma17040900_

Round 1

Reviewer 1 Report

Comments and Suggestions for Authors

The authors have presented a detailed study of the mechanical properties of TiNb steels.  Whereas the overall study is worthwhile the effect of Ti (Figure 4) does not appear to be large.  It may be possible that Ti addition can improve the cracking situation but the ductility data (Fig. 4) indicate the effect appears to be small.  It would be good if the authors could either recognise that the effect is small in the conclusion or make a clearer point regarding the Ti effect.  The temperature and strain rate dependencies are useful information and are worthy of publication.

Comments on the Quality of English Language

OK

Author Response

Dear Reviewer,

We are very grateful for your constructive suggestions and for pointing out confusing issues in the text.

Considering your valuable comments, we tried to thoroughly revise the manuscript and worked on almost all points stated in Review Report 1 as follows:

-----------------------------------------------------------------------------------------------------------------------

Reviewer #1
The authors have presented a detailed study of the mechanical properties of TiNb steels.  Whereas the overall study is worthwhile the effect of Ti (Figure 4) does not appear to be large.  It may be possible that Ti addition can improve the cracking situation but the ductility data (Fig. 4) indicate the effect appears to be small.  It would be good if the authors could either recognize that the effect is small in the conclusion or make a clearer point regarding the Ti effect.  The temperature and strain rate dependencies are useful information and are worthy of publication.

Response: Thanks for this feedback.

  • This is an editorial comment that has somehow been included-please delete.(Line 27-35)

Response: This problem has been solved.

  • Are these references? Just las name. If they are separate then identify which goes with each name. (Line (99)

Response: Yes, these are from the same reference. As required, this problem has been solved.

  • The legend is not clear enough- please make the colours/sizes larger or use symbols so the curves can be clearly matched to the legend.(Fig 3) (Line 163)

Response: MatCalc curves have been improved.

  • That is a stretch given the small differences shown. (Line 199-200)

Response: This sentence has been reformulated.

  • Please identify AF and PF in the caption. (Line 253-254)

Response: This problem has been solved.

  • Whereas the overall study is worthwhile the effect of Ti (Figure 4) does not appear to be large. It may be possible that Ti addition can improve the cracking situation but the ductility data (Fig. 4) indicate the effect is small.

Response: That is true. The conclusion part has been re-organized and this non-relevant point have been improved.

-------------------------------------------------------------------------------------------------------------------------

We look forward to hearing from you regarding our submission and to respond to any further questions and comments you may have.

Sincerely,

Serkan Turan

--------------------------------------------------------------------------------------------------------------------------

Reviewer 2 Report

Comments and Suggestions for Authors

In this manuscript, the author investigated the effects of reheating rate (during continuous casting cooling process), titanium content, and deformation rate on the hot ductility of Nb containing micro-alloyed steels. The research results provide good guidance for reducing the cracking problem of Nb containing micro-alloyed steel billets in actual production processes. The experimental part of the manuscript is sufficient and systematic, but further improvement is needed in writing. The main problems that exist are:

1.The first paragraph of the introduction should be deleted.

2.The introduction is too lengthy and not logical.

3.The analysis of the experimental results is insufficient and confusing. As shown in the analysis above Figure 4.

4.The conclusion is lengthy and unorganized.

5.There are many incomprehensible places in the manuscript. As shown in line 200 increasing Ti- containing steels were prone to reveal better hot ductility. And in line 316 The mean diameter size of crack induced particles varied approximately between 80 nm to 1 µm.

6.Some of the analyses in the manuscript are too far fetched. For example, in line 185, How did the author determine that the low temperature end of the ductility trough is 700 ℃?  And in line 325 it can also be claimed that fine NbC is .......

Comments on the Quality of English Language

There are many places where semantic representation is unclear

Author Response

Dear Reviewer,

We are very grateful for your constructive suggestions and for pointing out confusing issues in the text.

Considering your valuable comments, we tried to thoroughly revise the manuscript and worked on almost all points stated in Review Report 2 as follows:

-----------------------------------------------------------------------------------------------------------------------

Reviewer #2
In this manuscript, the author investigated the effects of reheating rate (during continuous casting cooling process), titanium content, and deformation rate on the hot ductility of Nb containing micro-alloyed steels. The research results provide good guidance for reducing the cracking problem of Nb containing micro-alloyed steel billets in actual production processes. The experimental part of the manuscript is sufficient and systematic, but further improvement is needed in writing. The main problems that exist are:

  • The first paragraph of the introduction should be deleted.

Response: That was an editorial comment and has been deleted as required.

  • The introduction is too lengthy and not logical.

Response: The introduction part has been reedited and improved.

  • The analysis of the experimental results is insufficient and confusing. As shown in the analysis above Figure 4.

Response: This paragraph under the subtitle of “thermomechanical tests” has been remodified and improved as much as possible.

  • The conclusion is lengthy and unorganized.

Response: The conclusion part has been reorganized and edited.

  • There are many incomprehensible places in the manuscript. As shown in line 200 ‘increasing Ti- containing steels were prone to reveal better hot ductility’. And in line 316 ‘The mean diameter size of crack induced particles varied approximately between 80 nm to 1 µm’.

Response: Related sentences have been revised.

  • Some of the analyses in the manuscript are too far-fetched. For example, in line 185, How did the author determine that the low temperature end of the ductility trough is 700 ℃? And in line 325 ‘it can also be claimed that fine NbC is ......’.

Response: 1) The main factor causing microcracks at the low temperature end of hot ductility is ferrite formation. Therefore, this explanation in line 185 has been given. Additionally, Table 2 has been added as reference at the end of this sentence.

2) Related sentences in line 325 have been improved.

-------------------------------------------------------------------------------------------------------------------------

We look forward to hearing from you regarding our submission and to respond to any further questions and comments you may have.

Sincerely,

Serkan Turan

--------------------------------------------------------------------------------------------------------------------------

Reviewer 3 Report

Comments and Suggestions for Authors

Dear Authors,
Please refer to the comments in the attached file.

Author Response

Dear Reviewer,

We are very grateful for your constructive suggestions and for pointing out confusing issues in the text.

Considering your valuable comments, we tried to thoroughly revise the manuscript and worked on almost all points stated in Review Report 3 as follows:

-----------------------------------------------------------------------------------------------------------------------
Reviewer #3

  • In lines 27-35 of the article, the authors cite several characteristics with which the abstract should be characterised. In an article submitted for publication in a journal of such renown, such a reference is, in the reviewer's opinion, unnecessary. Even perversely, the authors pointed out that the introduction should refer to other publications. In this article, the literature review is unacceptably poor. 16 literature items are far too few.

Response: In lines 27-35 given paragraph somehow from Reviewer added. This unnecessary part has been deleted. Literature has been enhanced. Additional literature references have been added at many points in the introduction part.

  • At line 40, the authors write: A series of studies indicated that the crack formation of steel slabs depends on a number of process parameters.... Please make a precise reference to the mentioned studies and indicate the literature items.

Response: Related missing references have been added.

  • In line 40 there is no space between the dot and the letter A.

Response: This problem has been solved.

  • Line 47-49: It can be generally stated that the ductility decreases particularly between 700 – 1000 °C, which makes the material susceptible to cracks as a result of exceeding stress level for endurance against the crack initiation- why such conclusions, please provide a literature reference or your own research

Response: References has been given relevantly.

  • Line 54- no full stop at the end of the task.

Response: This problem has been solved.

  • Line 121- no space between the full stop and the number 1.

Response: This problem has been solved.

  • Sloppy formation of text- in some places the word Fig bolded, in some places not.

Response: The words Fig. and Table have been changed to bold if they were not.

  • It is interesting to note how the tensile tests were carried out under a specific temperature regime- please could the authors include a photo of the test so that we have an overview of the test method and instrumentation used.

Response: New images (Figure 1) have been added.

  • Was and elongation sensor used in the tensile test?

Response: Experiments have been performed with push rod quenching and deformation dilatometer (Bähr Thermoanalyse DIL 805A/T). It consists of LVDT measuring system which is also measuring the elongation under thermomechanical treatment. Different strain rates, reheating rates and tem values could be programmed and tested in the tensile test in this study.

  • In the results section I am missing the presented results of the tensile test i.e. the nominal values of the mechanical characteristics of the materials determined and the presentation of the stress-strain relationship

Response: In this study, for each parameter minimum 2 experiments have been performed. That means we have more than 500 stress-strain curves totally. Instead of this, Figure 4 was presented which contains average values of the hot ductility (reduction of area) results obtained from these stress strain curves.

  • Line 203- what is the figure number?

Response: This problem has been solved.

  • Line 254- remove space

Response: This problem has been solved.

  • Line 275- double space in the text

Response: This problem has been solved.

  • For better readability of the article, I suggest introducing a space between the text and the figures.

Response: This problem has been solved.

  • Line 313- no space.

Response: This problem has been solved.

  • In the conclusion section, I suggest relating the research results obtained to those of other researchers.

Response: This suggestion could not be met because there was not sufficient data in the literature to meaningfully correspond to the steel grade and parameter combination investigated in this study.

  • Line 111- double paranthesis

Response: This problem has been solved.

  • Line 314- space between 2 and 20

Response: This problem has been solved.

-------------------------------------------------------------------------------------------------------------------------

We look forward to hearing from you regarding our submission and to respond to any further questions and comments you may have.

Sincerely,

Serkan Turan

--------------------------------------------------------------------------------------------------------------------------

Round 2

Reviewer 2 Report

Comments and Suggestions for Authors

Does Figure 6 indicate that cracks are more likely to appear around polygonal ferrite? Generally speaking, the strength of ferrite is much lower than that of austenite. The thinner the pre eutectoid ferrite film at the austenite grain boundary, the easier it is for stress concentration to occur, resulting in cracks.

Author Response

Dear Reviewer,

Please find our response letter in the attachment.

Kindly regards

Serkan Turan
